# Encapsulation of Phenolic Compounds Extracted from Beet By-Products: Analysis of Physical and Chemical Properties

**DOI:** 10.3390/foods13182859

**Published:** 2024-09-10

**Authors:** María del Cisne Guamán-Balcázar, Magdalena Montero, Alejandro Celi, Antonio Montes, Ceferino Carrera, Clara Pereyra, Miguel Ángel Meneses

**Affiliations:** 1Department of Chemistry, Universidad Técnica Particular de Loja, San Cayetano Alto, Loja 1101608, Ecuadoraiceli3@utpl.edu.ec (A.C.); mameneses@utpl.edu.ec (M.Á.M.); 2Engineering and Food Technology, Faculty of Sciences, University of Cádiz, 11510 Puerto Real, Spain; antonio.montes@uca.es (A.M.); clara.pereyra@uca.es (C.P.); 3International Excellence Agrifood Campus (CeiA3), 11510 Puerto Real, Spain; ceferino.carrera@uca.es; 4Department of Analytical Chemistry, Faculty of Sciences, IVAGRO, University of Cadiz, 11510 Puerto Real, Spain

**Keywords:** beet by-product, encapsulation, optimization, physical and chemical properties

## Abstract

Beet is a nutritious and health-promoting food with important bioactive compounds in its industrial by-products. The encapsulation of antioxidants from beet by-products has been proposed for valorization. For this, an ethanol–water extract was mixed with polyvinylpyrrolidone (PVP) (used as a carrier agent) and then encapsulated. The encapsulation was performed by spray drying, where the effects of temperature (140–160 °C), extract input flow rate (10–30%), and extraction solvent (ethanol–water 50/50 *v*/*v* and ethanol) were evaluated for the total phenol content and the spray-drying yield. The yields obtained were between 60 and 89%, and total phenols were between 136 and 1026 mg gallic acid equivalents/g of encapsulated product. Both responses were affected (*p* < 0.05) by the extraction solvent. The optimal spray-drying conditions were determined by response surface methodology (RSM). The encapsulated product obtained at optimal conditions was characterized by infrared spectrometry, X-ray fluorescence, Ultra-High Performance Liquid Chromatography, and scanning electron microscopy analysis. The results show that the encapsulated product has a high content of total phenols and compounds such as betanin, isobetanin, and neobetanin. Considering the results of physicochemical properties and the bioactive compounds, the optimized encapsulated product could be applied in the food industry as a bioactive ingredient or natural colorant. However, the further investigation of alternative carrier agents needs to be performed to reduce caking.

## 1. Introduction

Beetroot (*Beta vulgaris* L.) is a fairly complete food that contains no fat, is rich in fiber, and is effective against anemia and constipation [1]; it contains vitamins A and C, vitamin B6, niacin, biotin, and folic acid, in addition to minerals, such as iron, magnesium, selenium, potassium, calcium, zinc, phosphorus, and sodium, and phenolic compounds, such as phenolic acids, flavonoids, and organic and inorganic acids [2].

Beetroot sugar is essentially sucrose with certain amounts of fructose and glucose, which is important because fructose is not a sugar recommended by nutrition experts, especially for athletes [3]. It contains betalains, which are natural pigments that have nitrogen and are responsible for giving bright color to the fruits, flowers, and roots of the plants in Caryophyllales [4]. The phenolic compounds present in beet are natural antioxidants that defend the organism from microbial and infectious attacks and help to reinforce the immune system [5]. They also have functions of cell signaling, gene expression, and enzyme activity, which is another form of immune system protection [6]. In the food industry, due to beetroot’s nutritional and sensory properties, beetroot powder, extract, paste, or juice are used in the production of yoghurt, ice cream, beetroot jelly, beetroot candy, snacks, and biscuits fortified with red beetroot [7].

For greater use of the phytochemical properties of plant-based foods, the use of by-products as a source of antioxidant compounds is a sustainable alternative, especially when it has been reported that waste is around 20% in tubers, fruits, and vegetables, which during processing can represent losses of more than 30%. According to Eurostat [8], on the European continent, around 112 million tons of waste is produced per year as a result leaving a large proportion of usable raw materials. Different studies have determined that these by-products contain a greater amount of bioactive components than the processed part. Beet by-products such as peel, seeds, fiber, leaf, and roots are particularly rich in phenolic compounds, ascorbic acid, flavonoids, carotenoids, betalains, nitrate, saponins, ferulic acid, taurine, triterpenes/steroids, sesquiterpenoids, and alkaloids [7]. Therefore, a promising approach to utilizing these by-products is through the extraction and encapsulation of their active compounds.

The encapsulation of the active principle in a protective matrix is proposed to maintain and take advantage of the phytochemical properties of compounds of plant origin. The encapsulation increases the shelf life of the product, avoiding degeneration due to environmental factors such as light, O_2_, or pH, allowing the development of ingredients for the food and pharmaceutical industries with the possibility of preserving their bioactivity for a longer period. Spray drying is a classical technique used for encapsulation. It consists of dispersing a liquid solution of core and protective substances into a dryer gas at high temperature, and the rapid evaporation of liquid causes the core components to be covered in a protective layer, which is known as an encapsulant. This encapsulant protects the bioactive compounds and also prevents the sticking of formed particles to the internal walls of the drying chamber. Thus, a good encapsulant allows us to increase the process yield and to reduce the stickiness of the final product [9]. Several studies have determined the optimal processing conditions in spray drying to obtain beetroot juice-encapsulated powder. Several processing parameters have been evaluated, such as the type and amount of encapsulant (maltodextrin, gum arabic, inulin, protein isolate, pea protein), inlet drying gas temperature (125–180 °C), and feed flow rate (9–13 mL/min), on the physicochemical, structural, and functional properties of the powder [9,10,11,12,13,14,15].

Singh et al. [9] evaluated the effect of temperature, feed flow rate, and maltodextrin concentration on density (0.618–0.747 g/mL), moisture content (3.95–6.50%), and betalain content (12.03–38.52 mg/100 g of dry matter), determining that the optimal conditions were a feed flow rate of 10 mL/min, temperature of 149 °C, and maltodextrin concentration of 20%. Similarly, Bazaria et al. [14] determined the best conditions in terms of powder yield (41.31–54.63%), hygroscopicity (14.46–20.68 g of water/100 g of dry matter), redness value (22.56–34.67), betalain retention (61.34–70.46%), and radical scavenging activity (56.55–67.76%) to be an inlet air temperature of 160 °C, a feed flow rate of 400 mL/h, and a maltodextrin concentration of 15%.

Given the limited information on the use of beetroot by-products for producing powder via spray drying, this research aims to optimize the inlet temperature, feed flow rate, and extraction solvent to maximize both process yield and total phenol content as well as to measure the physicochemical properties of the optimized encapsulated powder. The results are considered important for the economic valorization of beet by-products.

## 2. Materials and Methods

### 2.1. Raw Materials

The beetroot (*Beta vulgaris*) samples were purchased at the wholesale market in Loja, Ecuador, in October 2020, and immediately transported to the laboratory. The samples were classified and washed using 5% sodium hypochlorite for 10 min, which ensured that bacteria and contaminating agents were eliminated.

The beetroot samples were crushed to separate the liquid (juice) from the solid (by-product, essentially exhausted fiber). The solid residue was dehydrated in a convective tray dryer (Lassele DY-110H, Ansan, South Korea) for 13 h at 60 °C. The dried by-product was then reduced in size using an ultra-centrifugal mill (Retsch Mill ZM200, Düsseldorf, Germany) and sieved (J. ENGELSMANN AG, Ludwigshafen, Germany) to obtain the appropriate particle size (250 to 125 µm). The by-product powder was packed in plastic–aluminum bags and stored at room temperature (18 °C) until further analysis.

### 2.2. Materials and Chemicals

Sodium hypochlorite (5%), 2,2-dyphenil-1-picrylhydrazyl (DPPH), Folin and Ciocalteu Phenol Reagent (2N), gallic acid, ±-6-hydroxy-2,5,7,8-tetramethylchromane-2-carboxylic acid (≥97%), 2, 4, 6-Tris (2-pyridyl)-striazine (≥98%), ferric chloride hexahydrate (≥98%) polyvinylpyrrolidone (MW 10000), acetic acid and acetonitrile were purchased from Sigma-Aldrich (Steinheim, Germany). Sodium carbonate, hydrochloric acid (37%), glacial acetic acid, and ethanol (99.8%) were purchased from Merck (Darmstadt, Germany), and sodium acetate trihydrate was purchased from Mallinckrodt Chemicals (Mexico, México).

### 2.3. Preparation of Beetroot By-Products Extracts

The extraction of antioxidant compounds was performed by dynamic maceration. For this procedure, 30 g of by-product powder was put in contact with 500 mL of solvent (ethanol 99.8% and ethanol–water 50/50 *v*/*v*, separately) in continuous stirring at 400 rpm for 4 h at 40 °C. Then, the liquid extracts were separated by filtration. Two extracts were obtained, ethanolic extract and ethanol–water extract. The ethanol in the ethanol–water extract was eliminated by vacuum evaporation to obtain an aqueous solution, while the ethanol extract was maintained as such. Both extracts were stored in refrigeration at 4 °C until its processing.

### 2.4. Spray Drying to Encapsulate Antioxidants from Beetroot By-Products

#### 2.4.1. Preparation of Sample

Prior to the spray-drying processing, the extracts were mixed with polyvinylpyrrolidone (PVP) and used as an encapsulant. The amount of PVP added was enough to adjust the concentration of total solids for each extract until reaching 5% (*w*/*v*).

#### 2.4.2. Spray-Drying Processing

Spray drying was performed in a Mini-Spray Dryer (BÜCHI B-290, Flawil, Switzerland). The extracts were fed (10% to 30% pump rate or 2.5 to 9 mL/min feed flow) through a 150 µm nozzle to the spraying chamber where a drying gas was flowing at 35 m^3^/h. Nitrogen (E941 99.998%, INDURA, Guayaquil, Ecuador) was used as the drying gas at temperatures between 140 and 160 °C (Figure 1). The solid product (encapsulated) collected in the separator vessel was packed in plastic–aluminum bags and kept in a desiccator chamber until the physico-chemical analysis was performed.

#### 2.4.3. Determination of the Spray-Drying Yield

The spray-drying yield was calculated considering the weight of the collected material (*final weight*) in reference to the weight of total contained solids in the initial solution (*starting weight*) as presented in Equation (1).
(1)$$Yield\;(\%)=\frac{final\;weight}{Starting\;weight}\times 100$$

### 2.5. Physical Properties Analysis

The physical properties analysis were carried out in both the dehydrated by-product and the spray-dried product.

#### 2.5.1. Moisture Determination

The moisture content was determined following the AOAC Standard 934.06 [16] procedure. A measured quantity of each sample was placed in an oven (Cole Parmer 52000-70, Chicago City, IL, USA) at 106 °C for 2 h. After this time, the samples were cooled in a desiccator for 20 min and weighed. This procedure was repeated until it reached constant weight.

Moisture content was calculated by weight loss using the following Equation (2):(2)$$Moisture\;Content=\frac{Starting\;weight-final\;weight\;}{Starting\;weight}\times 100$$

#### 2.5.2. Water Activity (a_w_)

The water activity measure was performed using water activity equipment (Lab touch aw Novasina, model NV260 0179, Lachen, Switzerland). A portion of each sample was placed in the sample holder of the equipment, and the following data were registered: water activity, temperature and elapsed time.

#### 2.5.3. Color

Color was registered in a CR-10 Tristimulus colorimeter (Konica Minolta Sensing Americas, Inc., Ramsey City, NJ, USA), using the Y, x, y system and then transforming these coordinates into the CIE color space, *L**, *a**, and *b**. The hue angle (*h*°) and chroma (*C**) were calculated with the following equations:(3)$$h\;{*}_{ab}=arctg\;\frac{b*}{a*}$$
(4)$$C^{*}={(a^{*2}+b^{*2})\;}^{1/2}$$

#### 2.5.4. Density

The density of solid samples was determined by the method reported by Shenoy et al. [17] with some modifications. Approximately 1 g of the encapsulated product was placed in a graduated cylinder (10 mL vol) and then hit with a bench 100 times at a height of 5 cm. Following this, the density was calculated by dividing the mass by the occupied volume in the cylinder.

#### 2.5.5. Hygroscopicity

The hygroscopicity was determined through a gravimetric method as reported by Oliveira Menezes et al. [18] and the GEA Niro Research Laboratory [19] with some adaptations. First, 2 g was put into an airtight flask and introduced inside an incubation chamber at constant temperature and saturated relative humidity (45 °C and 65% relative humidity). The weight of the samples was controlled based on the gain of weight until the result was stable. The result was expressed in % humidity according to Equation (5):(5)$$Hg=\frac{m_{f}-mm}{mm\times (1-x_{w})}\times 100$$
where *Hg*: hygroscopicity; m_f_: final sample weight; *mm*: original weight of the sample; and *x_w_*: initial sample moisture.

#### 2.5.6. Degree of Caking

The degree of caking was determined after the hygroscopicity analysis. Once the sample was hydrated, it was placed in an oven at approximately 102 °C for one hour. It was dried, cooled, then weighed and sieved to 0.5 mm mesh. The sieve had constant agitation (5 min at 50 Hz) [19]. The mass of powder retained in the sieve meshes was registered, and the degree of caking was calculated through Equation (6):(6)$$\%DC=\frac{c}{d}\times 100\%$$
where *c*: mass of the sample retained on the sieve and *d*: mass of the initial sieved sample.

#### 2.5.7. Dispersibility

Dispersibility was determined according to the methodology used by Abdalla et al. [20] with some modifications. Briefly, 1 g of encapsulated powder was added to 10 mL of distilled water at 25 °C in a 50 mL beaker and stirred (1050 rpm for 5 min). The optical density of the supernatant was measured at 760 nm wavelength in a UV-visible spectrophotometer. Dispersibility was expressed in terms of optical density (OD) units.

#### 2.5.8. Morphology of Encapsulated Product

The morphology and particle size of the encapsulated powder were evaluated using a Nova NanoSEMTM 450 (Elecmi, Zaragoza, Spain) scanning electron microscope. Before the analysis, the samples were covered with a 15 nm gold film using a sputter coater (The Cressington Scientific Instruments Ltd., Watford, England). To determine the particle size, SEM images were processed using ImageJ software (Version 1.54).

### 2.6. Chemicals Analysis

#### 2.6.1. Determination of Total Phenols

The total phenol content was measured by the Folin–Ciocalteu method, as reported by Thaipong et al. [21]. First, 150 µL of the sample solution was mixed with 150 µL of 0.25 N Folin–Ciocalteu solution and 2400 µL of distilled water; this mixture was stirred for 2 min and then allowed to react for 3 min. After that, 300 µL of 1N sodium carbonate was added and left to react for 2 h in the dark at room temperature. The absorbance of each sample was measured at 725 nm wavelength. Gallic acid was used as standard at concentrations between 0 and 100 mg/L. The results were expressed as mg of gallic acid equivalents per grams of encapsulated product (mg GAE/g encapsulated).

#### 2.6.2. Infrared Spectroscopy (FTIR)

For the development of this method, the Nicolet iS10 Mid Infrared FT-IR spectrometer was used with a 120 mW cm^−2^ UV laser accessory (Power Arc UV 100 Blue Sky Special Lamps Development Co., Ltd., Zhuozhou, China). Using the main software of the equipment, the complete recognition of each compound was carried out through an FTIR spectrum.

#### 2.6.3. X-ray Fluorescence (XRF)

An XRFp spectrometer and IPAQ system were used, and the Bruker S1 program was selected at the beginning. For the measurements, the Mining Light Elements FP method was established for major compounds > 1% and minor compounds 1–0.1%, while for traces, the Soil FP method was applied (<0.1%). For this purpose, a sample of 2 g was placed on the sample holder and covered; this was focused with the X-ray beam, and the sample holder was covered with the lead shielding. The sample was irradiated with X-rays, and the data were recorded.

### 2.7. Betalains Determination

To determine and quantify the betalains present in the encapsulated product, 100 mg of the sample was dissolved in 25 mL of deionized water. The total betalain content was determined via colorimetric analysis. The encapsulated sample was diluted with deionized water to achieve appropriate absorption values. Absorbance was measured at 540 nm using a Cary 60 UV-Vis spectrophotometer (Agilent Technologies, Santa Clara, CA, USA). The total betalains content was calculated using the equation proposed by Cai and Corke [22]:(7)$$Total\;Betalains\;Content\;\left(mg/g\right)=\frac{A\times V\times DF\times MW\times 10^{-3}}{\epsilon \times L\times W}$$
where *A* is the absorbance at 540 nm, *V* is the volume of the sample (L), *DF* is the dilution factor, *MW* is the molecular weight of betanin (550 g/mol), ϵ is the molar extinction coefficient (65,000 L/mol·cm), *L* is the path length of the cuvette (1 cm), and *W* is the weight of the sample.

The separation and quantification of betalains was performed using an Ultra-High Performance Liquid Chromatography (UHPLC) system coupled with a photodiode array (PDA) detector (ACQUITY UPLC^®^ H-Class, Waters Corporation, Milford, MA, USA). A UPLC^®^ BEH C18 column (Waters Corporation, Milford, MA, USA; 100 × 2.1 mm I.D., particle size 1.7 µm) was used with the oven temperature maintained at 37 °C. The mobile phase consisted of a binary solvent system: phase A was a 2% acetic acid solution in water, and phase B was a 2% acetic acid solution in acetonitrile. The flow rate was set at 0.55 mL/min. The gradient program over 9 min was as follows (time, % solvent B): 0 min, 0%; 1 min, 0%; 3 min, 5%; 4 min, 10%; 4.5 min, 10%; 5 min, 20%; 7 min, 20%; 8 min, 30%; 9 min, 100%. Before the chromatographic analysis, extracts were filtered through a 0.2 µm nylon syringe filter (Membrane Solutions, Dallas, TX, USA). The injection volume was 3 µL. All analyses were performed in duplicate. 

### 2.8. Experimental Design and Statistical Analysis

In order to evaluate the effect of the spray-drying operating conditions on the encapsulation yield and the total phenols content, a multilevel 2^3^ factorial design was selected. The factors are listed in Table 1. Analysis of variance (ANOVA) and the least significant difference test (*p* < 0.05) allow us to determine the significance of the factors over the responses. The response surface methodology (RSM) was applied to determine the operative conditions that maximize the encapsulation yield and the total phenols content. Statgraphics Centurion XVI.I software was used in the statistical analysis.

## 3. Results and Discussion

### 3.1. Characterization of Beetroot By-Product

Considering water as the main component of various foods, moisture analysis is fundamental, since it directly interferes with biochemical and enzymatic reactions as well as with the interaction with its own components and microorganisms or those foreign to the food. Table 2 shows the moisture, water activity and color of the dehydrated beetroot by-product. Regarding humidity, the value obtained was 6.21 ± 0.08%.

Regarding water activity (WA), as in the case of humidity, it is related to the water content of the food. However, WA refers to the amount of bound or free water in it. Shelf life is thus considered essential in terms of microbial activity [23]. The value of the beet by-product was 0.30 ± 0.00, so the product is stable against microbial growth, since microorganism growth is not expected at values below 0.6 ± 0.00.

Concerning color, *L** represents the lightness that goes on a scale from black to white, *a** represents colors that go from green (−) to red (+), and b* indicates colors that go from blue (−) to yellow (+). These values are used to define *C** (Chroma*), which is an indicator of color saturation. Likewise, the *h°* angle “is the angle that measures the hue, indicating the relative orientation of the color with respect to the 0° origin” [24].

The data obtained show *L**: 34.3 ± 0.4 is not high; *a**: 20.3 ± 0.1 is a positive value that tends toward red colors or reddish tones; and *b**: 3.5 ± 0.5 tends to yellow colors, which although being a low value, it cannot be considered a dominant color for beets. *C**: 20.38 ± 0.32 indicates a low color saturation level and is oriented toward red colors. For the hue angle represented by *h°*: 80.27, shades ranging from red to yellow are identified.

The measurements of color in raw materials as in the case of dehydrated beet by-products could be an indicator of the quality and persistency of betanins after the post-harvesting processing. This means that the dehydration temperature, the milling and storage steps do not completely decrease bioactive compounds in the beet by-product powder.

### 3.2. Encapsulation of Antioxidants from Beetroot By-Products

The effects of solvent composition in the extraction process, temperature and extract flow in the spray-drying process over the encapsulation yield and total phenols content were determined based on analysis of variance (ANOVA), while a regression analysis was used to optimize both responses, demonstrating the empirical relationship between the factors studied and the responses.

The Pareto chart (Figure 2) indicates the statistical significance of simple effects and interaction of the factors studied on the encapsulation process. Accordingly, the encapsulation yield was affected (*p* < 0.05) only by the extract solvent (x3) (Figure 2a), while the extract solvent (x3) and the extract flow (x2) influenced (*p* < 0.05) the total phenols content (Figure 2b).

The selected design produced a mathematical model (Equations (8) and (9)) that determines the encapsulation yield (R^2^= 72.06%) and total phenols content (R^2^ = 99.90%) as a function of factors x1, x2 and x3 and their interactions within the levels shown in Table 1.
(8)$$\begin{array}{cc} Yield(\%)= & 51.9072+0.170033\times Temperature+1.54943\times Extract\;flow-24.8274\times Extract\\ & -\;Solvent-0.00891333\times Temperature\times Extract\;flow\\ & +\;0.200951\times Temperature\times Extract-Solvent\\ & -\;0.0861787\times Extract\;flow\times Extract-Solvent \end{array}$$
(9)$$\begin{array}{cc} Total\;phenolic & \left(mg\frac{GAE}{g}encapsulated\right)\\ & =\;430.217+0.997789\times Temperature+0.94144\times Extract\;flow+314.499\times Extract\\ & -\;Solvent-0.0135257\times Temperature\times Extract\;flow\\ & +\;0.753465\times Temperature\times Extract-Solvent-0.755065\times Extract\;flow\times Extract\\ & -\;Solvent \end{array}$$

### 3.3. Optimization of the Influence of the Factors

According to the response surface methodology, Figure 3a,b present the results for the encapsulation yield and total phenols content, respectively. Both figures were set for a middle constant value of x2 (temperature, 150 °C), and the changes in y (encapsulation yield and total phenols content) correspond to the effect of x1 (extract solvent) and x3 (extract flow). In Figure 3, it can be observed that increasing the ethanol/water solvent factor allows increasing the encapsulation yield and the content of phenolic compounds in the encapsulated product if compared to only ethanol as solvent. Considering this, optimization could be achieved by increasing the amount of water in the solvent. According to Fernando et al. [25], the extraction of antioxidant compounds is improved when ethanol or methanol is mixed with water, which occurs due to the polarity of the main beetroot compounds such as betalains. In addition, when ethanol and water are used as the solvent, the obtained products could be used as a natural colorant in the food industry. In this study, mixtures of ethanol and water were preferred as the solvent in accordance with the proposal of valorization through safe human consumption of the encapsulated antioxidants. The use of methanol is prohibited to avoid health risk if there is the presence of solvent in the encapsulated product as well as safety limitations in processing.

In the RSM, it was observed that the optimum value of % ethanol in water was at the extreme 50/50 (*v*/*v*) of the ethanol/water factor range. Therefore, the range was extended out using a higher percentage of water (60%, 70%, 80%, 100%) to maximize the encapsulation yield and total phenols content in the encapsulated powder.

In Figure 4a, it can be observed that by decreasing the ethanol concentration to 10% and 0%, the yield of the encapsulated powder decreases significantly (*p* < 0.05) compared to using 50%, 40%, 30% and 20% ethanol. On the other hand, with respect to the total phenols content (Figure 4b), the highest values were obtained when using 50%, 30% and 10% ethanol (*p* < 0.05). Therefore, considering the similarity in the results, ethanol–water 50/50 *v*/*v* was selected as the best extraction condition due to the ease in processing and to avoid the possible spoiling of liquid extracts prior to the encapsulation. The selection of the extraction technique significantly influenced the overall extraction yield. While ultrasound-assisted and microwave-assisted extractions have demonstrated efficacy in certain applications, dynamic maceration offers distinct advantages for large-scale operations. Its versatility and ease of scale-up make it an attractive option for the industrial-scale extraction of phenolic compounds. Dynamic maceration enables efficient extraction through the constant agitation of the sample in a solvent, facilitating mass transfer and enhancing the recovery of intracellular compounds. Moreover, the ultrasonic probe could be adapted to maceration equipment.

The optimal parameters to maximize the encapsulation yield and total phenols content in the encapsulated powder are shown in Table 3.

### 3.4. Physical Properties of Encapsulated Product

The processing conditions that maximize the encapsulated yield and total phenols content were selected from the RSM. The analysis of physical properties was performed on an encapsulated product obtained at the optimal conditions for encapsulated yield (Table 3). These results could be considered a characterization of the encapsulated ingredient.

Table 2 shows the physical properties for the optimal encapsulated product, such as color, water activity, humidity, density, solubility, dispersibility, hygroscopicity and degree of caking.

#### 3.4.1. Color

Regarding color, Table 2 shows the results obtained for *L** (51.02 ± 0.03) and *a** (35.36 ± 2.21), where a positive *a* value indicates tendencies toward red tonalities. Finally, the parameter *b** (6.67 ± 0.63) indicates tendencies toward yellow or blue tonalities, where a positive values indicates tendencies toward yellow.

The *h** section indicates the orientation of the color; the measurements of *h** (10.69 ± 0.34) obtained confirms the tendency to red tonalities. On the other hand, the *C** factor analyzed results (35.99 ± 2.29) indicate the quality of a medium-saturated color.

Based on the color characteristics (*L**, *a*, *b*, *C**, h*), the encapsulated product obtained indicates a medium bright color by the quality *L**; the factor a* reaffirms the tendencies to red color, which is far from the green tonality that would be the opposite; and finally, there is a slight tendency toward yellow tonalities, being the weakest characteristic. According to Nemzer et al. [26], the characteristic color of beetroots depends largely on the betacyanin/betaxanthin concentration ratio. The decrease in the red color in the encapsulated product compared to the initial dehydrated raw material corresponds to the encapsulating layer of PVP over the antioxidants. Depending on the quantity of encapsulant used, the measurement of color would be an indication of the effectiveness of the spray-drying process to produce encapsulated products.

#### 3.4.2. Moisture and Water Activity (WA)

Moisture content is a fundamental property, along with water activity, to determine the stability and storage of the encapsulated product [27,28]. In this context, Tontul and Topuz [29] mention that for powders, moisture values lower than 5% could be classified as microbiologically safe and could be stored for long periods of time. Likewise, as well as several physical properties, the moisture and physical state of a powder can be correlated with the efficiency at which a powdered or spray-dried food is completely reconstituted after prolonged periods of time. Table 2 shows the percentage of moisture obtained: 6.01 ± 0.28%. Similar results were reported in encapsulated products of beetroot juice (3.95–6.5%) [9], tamarind pulp (3.65–7.11%) [28], black mulberry juice power (8–16%) [30] and Juçara pulp (4.94–10.17%) [31]. However, for spray-dried beetroot juice powder, Carmo et al. [10] reported values ranging from 3.33% to 4.24% using maltodextrin, inulin, whey protein isolate, and their mixtures. In contrast, Bazaria et al. [13] determined values between 2.81% and 4.27% when using whey protein concentrate and temperatures between 160 and 180 °C. These differences in moisture content are primarily due to the type of encapsulant used and the drying temperature applied during the spray-drying process.

Regarding water activity (WA), as in the case of moisture, it is related to the water content of a product. Specifically, WA refers to the amount of bound or free water, which gives rise to the different chemical and biochemical reactions that can occur in a product. Therefore, this analysis is considered fundamental in terms of microbial activity and shelf life. Accordingly, the WA determined for the encapsulated product (0.19 ± 0.01, Table 2) is in zone I (WA: 0–0.25); thus, there is no probability of microbial growth [32]. Likewise, Tontul and Topuz [29] and Pereira et al. [31] mention that for values lower than 0.3, the product could be considered microbiologically and chemically safe.

In research related to beetroot juice powder obtained through spray drying, similar and even higher values have been reported compared to those obtained in the present research. For example, Gawałek et al. [12] determined values ranging from 0.18 to 0.23 using potato maltodextrin as the encapsulant. On the other hand, Janiszewska et al. [15] obtained values between 0.44 and 0.75 when using maltodextrin and between 0.44 and 0.75 when using gum arabic as the encapsulant.

#### 3.4.3. Density, Solubility and Dispersibility

Using the gravimetric method, the tapped density value was determined in the encapsulated antioxidant beetroot by-product (0.42 ± 0.03 g/mL). Higher values were reported in beet juice foam dehydrated with maltodextrin (0.88 ± 0.02 g/mL) [29] and beet powder dehydrated using the InfriDri^®^ process (0.753 g/mL) as well as lower values for powdered beet dehydrated using hot air (0.51 g/mL) [33]. Other studies report the bulk density; Bazaria and Kumar [13] presented values between 0.489 and 0.588 g/mL in spray-dried beet juice with whey protein concentrate; Fazaeli et al. [30] reported the density for black mulberry juice powder (0.3 to 0.5 g/mL); while Pereira et al. [31] reported lower values in Juçara pulp (0.14 to 0.37 g/mL).

As mentioned by Mimouni et al. [34], the density in powdered products can be strongly related to the speed and efficiency of dissolution in an aqueous medium as well as a measure of quality control against possible adulterations or modifications. Similarly, in the case of encapsulated powders, the type and percentage of encapsulant used will directly influence the density of the final product.

With respect to solubility, the obtained value was 84.47 ± 4.36%; this property is a quality factor in powdered substances and should be as high as possible in the solvents where it is solubilized. Similarly, in spray-dried products, the encapsulant used could help to increase this property. Similar values (83.15 ± 0.05%.) were obtained by Jelena et al. [35] in spray-dried beetroot juice with pumpkin protein isolate as well as in mango powders encapsulated with maltodextrin (90%) [36], while García-Segovia et al. [11] obtained lower values in beet juice atomized with pea protein (17.4–17.9%).

On the other hand, the dispersibility in the encapsulated antioxidants beet by-product was 0.95 ± 0.02 OD. This property measures the efficiency of a powder to solubilize as independent solid particles in aqueous medium; i.e., it measures the stability of a powder–aqueous medium dissolution. This property depends on the chemical composition, particle size, and density, among other physical properties [37].

The use of PVP as the encapsulant agent allows to increase the solubility and dispersibility of encapsulated products. PVP is known to favor the solubility of substances with low solubility in aqueous medium.

#### 3.4.4. Hygroscopicity and Degree of Caking

Hygroscopicity is the ability of a material to absorb water in the form of moisture during its storage. If a powder has low hygroscopicity, it can be considered a good pulverized material [28]. In this regard, hygroscopicity with a value of 15.14 ± 0.94% was obtained in the present study. Moisture content is directly related to hygroscopicity, i.e., a powder with low percentages of moisture has a greater capacity to absorb water from its environment and therefore holds greater hygroscopicity. However, this does not occur in all cases. In our case, a moisture percentage lower than 10% was determined as well as a low percentage of hygroscopicity (15.15%). However, due to the presence of beet sugars in the encapsulated product, there is no guarantee that this percentage will not increase with time despite the use of polyvinylpyrrolidone as the encapsulating agent. According to Caparino et al. [38], encapsulants could decrease hygroscopicity, protecting the product from the environment and improving storage stability. Hygroscopicity values of 16.5 ± 0.06% were reported for spray-dried mango powders by Caparino et al. [38]. In another study, a value of 30 gw/100 g dry solid was reported in orange juice powder; this value increased to 70 gw/100 g dry solid after 7 days [39], while Pereira et al. [31] reported a value of 12.26 ± 0.14% for Juçara pulp (*Euterpe edulis* M.). The difference in hygroscopicity values depends on the composition of the dehydrated food, the encapsulant matrix, and the drying technology used. The spray-drying process applies high temperature, and the final moisture is low while microparticles with a high contact surface are formed.

Regarding the degree of caking, it is an undesirable reaction in which powder is transformed into an agglomerated and sticky material due to the accumulation of plasticizing water on the surface of the powder, which causes a loss in the quality and functionality of the product. In this regard, due to the high sugar content in the beetroot (7.96 g/100 g vegetable) [40], the spray-dried powder had 100% caking. According to the GEA Niro Research Laboratory [19], which characterizes the hygroscopic behavior and the degree of caking, the encapsulated powder is classified as an extremely caking powder and caking product.

#### 3.4.5. Morphology of Encapsulated Beetroot By-Product Antioxidants

The morphology and microstructure of the beet by-product encapsulated powders can be observed in Figure 5. The beet by-product particles are spherical agglomerates in shape with an average size of 8.24 ± 5.92 μm. According to Biswas et al. [41], the morphology and shape of spray-dried nanostructured microcapsules can be controlled by adjusting the drying temperature. At lower temperatures, the particles are smaller and form spherical granules, while at higher temperatures, the tendency is to form hollow, larger particles. In this context, considering that in spray-drying encapsulation, it is rare to obtain particles smaller than 100 µm unless several optimizations are made [42], the obtained size (8.24 µm) can be considered small particles.

The results of physical properties provide valuable information about encapsulated products that is necessary to scale up to an industry level, including product development, equipment design or calculations, among others, being necessary during the processes of handling, transformation, formulation, and/or storage [43].

### 3.5. Chemical Properties of Atomized Beet By-Product Powder

#### 3.5.1. Infrared Spectroscopy (FTIR) 

Figure 6 shows the spectra corresponding to the atomized extract of beet by-product (1) and PVP (2). The different compounds of the atomized beet extract are represented by the letters A, B, C, D, E, F and G.

In the first instance, within the range 3200–2500 cm^−1^, there are values of 2925.75 (A) probably belonging to a band linked to the C-H sp3 stretching bond, while the value 3286.33 (B) represents a very broad and high band in the range 3500–3200 cm^−1^ corresponding to an O-H bond of water [44]. On the other hand, there are no bands in the 2500–2000 cm^−1^ region. However, in the 1500–600 cm^−1^ region, called the fingerprint, we found high peaks 1045.68 (C) in the 1100–1000 cm^−1^ ranges corresponding to carbohydrates and 1105.07 (D), which is a peak that represents the specific characteristics of the functional group and can change position based on the specific type of carbohydrate in the powder. In the case of PVP, no peaks are observed at the C and D positions. The peaks at 924.05 cm^−1^, 831.7 cm^−1^ and 831.21 cm-^1^ (E), at 1500–700 cm^−1^, are related to the presence of sucrose [45]. According to characterization and quantification studies of betacyanin pigments in several amaranth species, peaks between 1720 and 714 cm^−1^ correspond to compounds such as betacyanins [44].

Finally, the band within the range of 1500–1100 cm^−1^, specifically at 1291.32 cm^−1^ (F) and 1422.08 cm^−1^ (F), according to Mondragón [44], represents single bonds (C-C, C-N, C-O). In the range of 1700–1600 cm^−1^, representations of double-bond vibrations C=C and C=O are found. Regarding the peaks identified as G, 1652.09 cm^−1^ and 1648.19 cm^−1^, these may belong to the amide group due to their proximity to the 1650 cm^−1^ region, which is a characteristic of the (C=O) group. This is consistent with the presence of PVP (N-vinylpyrrolidone), which consists of a five-membered lactam attached to a vinyl group with the lactam being a cyclic amide.

#### 3.5.2. X-ray Fluorescence (XRF)

Information on the majority and minority compounds was collected through the Mining Light Elements method, and the soil method was used to determine the trace portion. Majority compounds included Al_2_O_3_ (0.787 ± 0.5575%), SiO_2_ (1.55 ± 0.2405%), P_2_O_5_ (0.93 ± 0.0612%), S (0.163 ± 0.0127%), K_2_O (3.11 ± 0.0245%), CaO (0.272 ± 0.0304%), and Fe_2_O_3_ (0, 435 ± 0.0014%), while in the trace compounds, we found ZnO (0.004 ± 0.0005%), MnO (0.024 ± 0.0015%) and Br (0.01 ± 0.005%).

Likewise, the analysis of the PVP-encapsulating agent was carried out. Among the main results, we found the presence of compounds such as SiO_2_ (1.7 ± 0.2135%), as the majority compound, and among the traces, compounds such as P_2_O_5_ (0.0738 ± 0.0387%), S (0.0775 ± 0.0106%), K_2_O (Not determined), CaO (0.075 ± 0.0042%), Fe_2_O_3_ (0.0198 ± 0.0013%) and ZnO (0.0035 ± 0.0009).

We can highlight that in the characterization of the atomized beet by-product, the contribution of SiO_2_ and ZnO is largely due to the presence of the encapsulant (PVP). In terms of food applications, components such as Al_2_O_3_, K_2_O, MnO and Br are typical of the beet by-product and correspond to the content of Al, K and Mn. In the determined quantities, they do not represent a hazard when consumed. In addition, Al has low intestinal absorption and would not represent a competition with antioxidant compounds.

### 3.6. Betalains

The total betalain content obtained for this sample was 5.571 ± 0.034 mg/g. Additionally, the UHPLC analysis revealed three major betalains: betanin, isobetanin, and neobetanin, representing 44.82%, 47.30%, and 3.76% wt. of the total betalains in the sample, respectively. Figure 7 shows the chromatogram obtained.

Some results were reported for betalains in beetroot juice powder obtained by spray drying, using maltodextrin as an encapsulant (12.02 to 367.60 mg/100 g d.b) [9,10,45], inulin (348.79 mg/100 g d.b.) [10], whey protein isolate (369.32 mg/100 g d.b) [10], whey protein concentrate (264.13 to 272.54 mg/100 g) [13], and arabic gum (129 mg/100 g d.m) [15]. It is important to consider that the contents of betalains found by these authors are lower than those of the present study, which is probably due to the different drying conditions used in the studies. According to Singh et al. [9], the higher values in betalains in the spray-drying process are mainly influenced by the process temperature, the encapsulant used, and the feed flow rate.

## 4. Conclusions

Phenolic compounds were effectively extracted from beet by-product using a ethanol–water mixture (50/50 *v*/*v*), and the spray-drying process was optimized in terms of encapsulation yield and total phenols content. The extraction solvent factor influenced both the encapsulation yield and total phenolic content, while the extract flow only affected the total phenolic content. The encapsulation of antioxidants from beet by-products was successfully optimized; in order to maximize encapsulation yield, the optimal conditions were as follows: temperature: 160 °C; extract flow: 30%; and solvent composition: 50% ethanol, while for total phenols content (mg GAE/g encapsulated), the optimal conditions were as follows: temperature: 160 °C; extract flow: 10%; and solvent composition, 50% ethanol, which were successfully defined in the encapsulated powder. Regarding the characterization of the encapsulated product, the stability against microbial growth was determined due to the low content of water activity and humidity; likewise, it is highly soluble in aqueous medium and presents low hygroscopicity. Nonetheless, the degree of caking is 100%, which could be an issue to consider for the correct storage of the ingredient for long periods of time in humid environments. Despite this, the elimination of sugars from the beet by-product extract as well as the use of another encapsulant could help to reduce caking. As for the chemical characterization, the presence of a peak of protein origin due to the presence of the encapsulating agent is highlighted as well as a peak characteristic of water and peaks that characterize the carbohydrates that make up the beet. In regard to the fluorescence spectrum, the presence of the K_2_O compound of the product and SiO_2_ that has its origin in the encapsulating agent are highlighted. However, these compounds do not represent a risk for health nor for its bioavailability. These results of the extraction and spray-drying process indicate that beet by-products can be valorized providing an encapsulated product with a good yield of antioxidant pigments, which could be used in the development of functional products such as ingredients or natural colorants with a contribution of phenolic compounds. Although the encapsulation efficiency has not been directly evaluated in this study, a high content of beet by-product extract inside the microencapsulates can be inferred from the results of betaine content and physicochemical characterization. In addition, further studies are needed to establish the stability of the powder during storage, as well as the bioavailability of the compounds present, to provide more comprehensive information on the potential health benefits of the beet powder.

## Figures and Tables

**Figure 1 foods-13-02859-f001:**
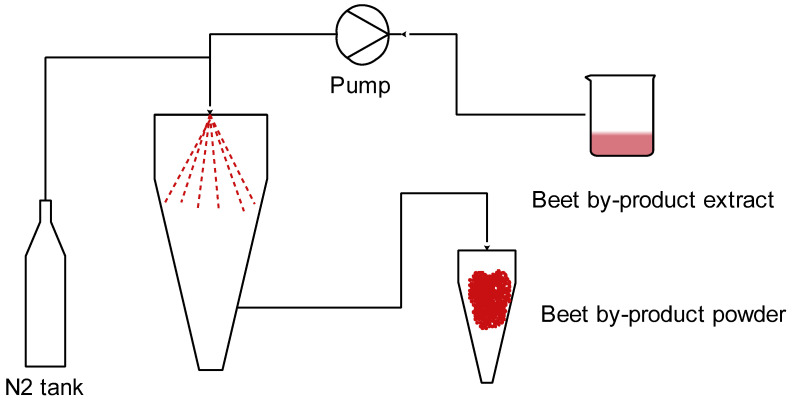
Beet by-product extract spray-drying process.

**Figure 2 foods-13-02859-f002:**
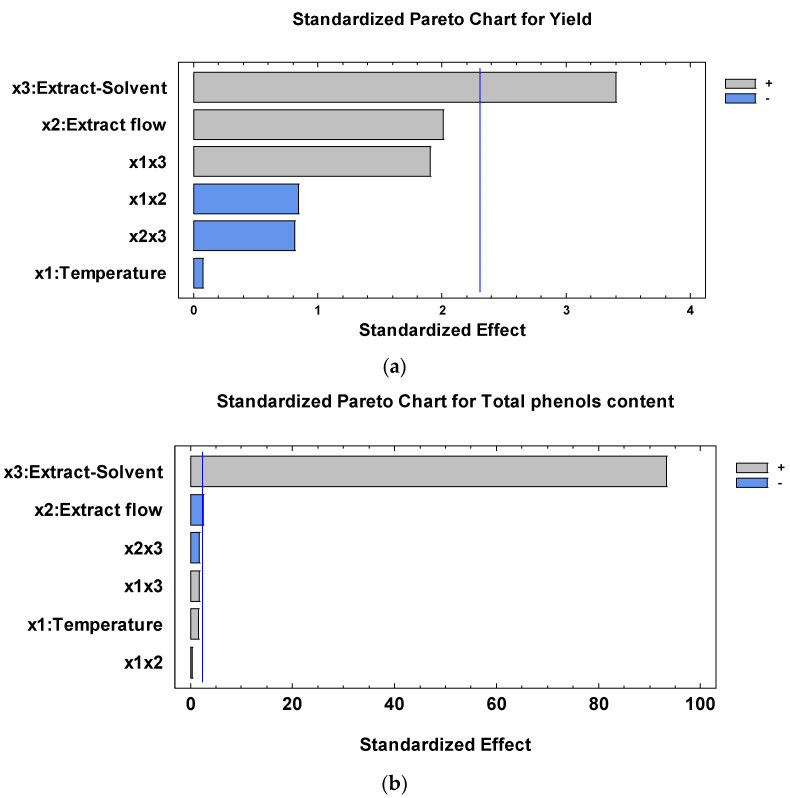
Pareto charts for the standardized effects of spray dryer factors on (**a**) encapsulation yield and (**b**) total phenols content.

**Figure 3 foods-13-02859-f003:**
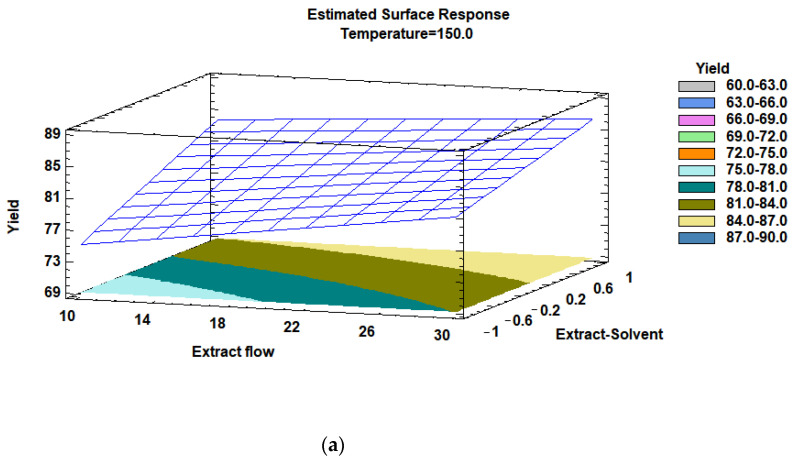
Response surface and contour plots displaying effects of encapsulation factors on (**a**) the encapsulation yield (%) and (**b**) the total phenols content (mg GAE/g encapsulated).

**Figure 4 foods-13-02859-f004:**
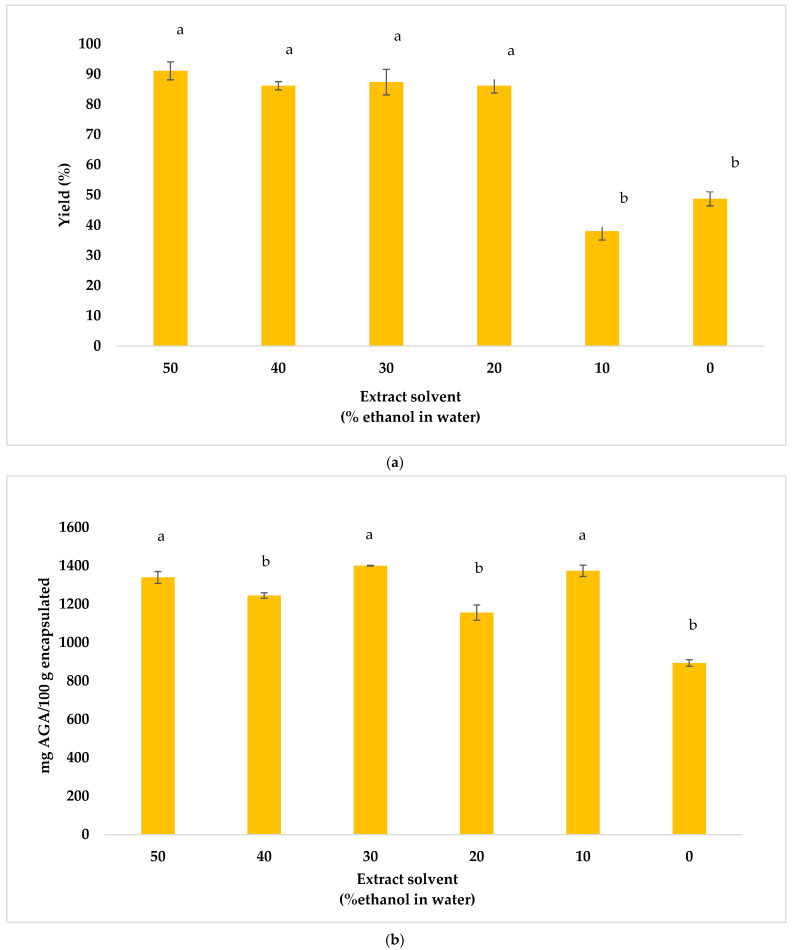
Optimization of extract solvent (percentages of ethanol in water), temperature = 160 °C and extract flow = 10%. (**a**) Yield; (**b**) total phenol content. Different letters in the bars mean significant differences (95% confidence level).

**Figure 5 foods-13-02859-f005:**
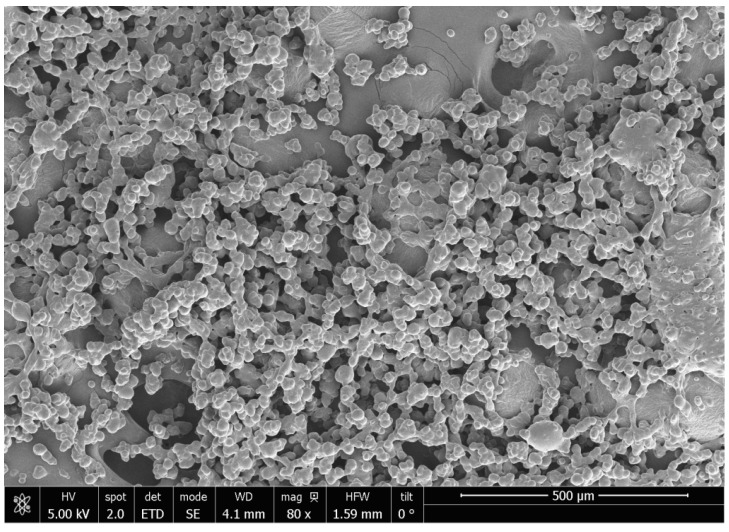
SEM images of beet by-product powder by spray dryer process.

**Figure 6 foods-13-02859-f006:**
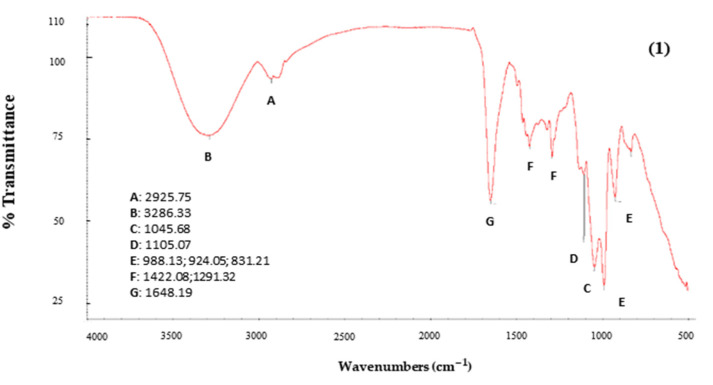
Infrared spectra. (**1**) atomized extract of beet by-product; (**2**) PVP.

**Figure 7 foods-13-02859-f007:**
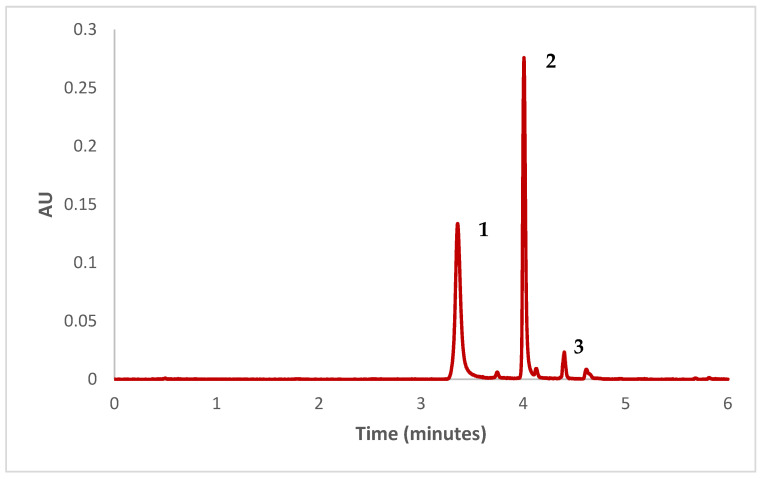
Chromatogram of the betalains identified in encapsulated (λ = 540 nm). 1. Betanin; 2. Isobetanin; 3. Neobetanin.

**Table 1 foods-13-02859-t001:** Experimental factors and levels in spray drying process.

Spray Dyer Factors	Levels		Unit
	−1	+1	
x1, Temperature	140	160	°C
x2, Extract flow	10	30	% pump capacity
x3, Extract-Solvent	50	100	% ethanol in water

**Table 2 foods-13-02859-t002:** Physical properties of beetroot by-product and encapsulated antioxidants.

Physical Property	Dehydrated By-Product	Encapsulated Product
Color		
*L*	34.3 ± 0.4	51.02 ± 0.03
*a**	20.3 ± 0.1	35.36 ± 2.21
*b**	3.5 ± 0.5	6.67 ± 0.63
*h**	80.68 ± 0.58	10.69 ± 0.34
*C**	20.38 ± 0.32	35.99 ± 2.29
Water activity	0.30 ± 0.00	0.19 ± 0.01
Moisture (%)	6.21 ± 0.08	6.01 ± 0.28
Density (g/mL)	-	0.42 ± 0.03
Solubility (%)	-	84.47 ± 4.36
Dispersibility (DO)	-	0.95 ± 0.02
Hygroscopicity (%)	-	15.14 ± 0.94
Degree of caking (%)	-	100 ± 0.00

Data are presented as mean and standard deviation, n: 3.

**Table 3 foods-13-02859-t003:** Optimized operative conditions that maximize the encapsulation yield and the total phenols content.

Spray Dryer Factor	Conditions	
	Yield	Total Phenols Content
Temperature, °C	160	160
Extract Flow, %	30%	10%
Extract–Solvent, % ethanol in water	50	50

## Data Availability

The original contributions presented in the study are included in the article, further inquiries can be directed to the corresponding author.

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
