# Peer review of "Encapsulation of Phenolic Compounds Extracted from Beet By-Products: Analysis of Physical and Chemical Properties"

_foods, 2024, doi:10.3390/foods13182859_

Round 1

Reviewer 1 Report

Comments and Suggestions for Authors

The manuscript you provided addresses several aspects related to food safety, specifically focusing on the impacts of biological, chemical, or physical hazards in food production or processing environments.

The manuscript presents a thorough investigation into the valorization of beet by-products through the production of an atomized powder rich in antioxidants, specifically focusing on the effects of spray drying parameters on the total phenols content and yield. The study addresses a significant issue in food processing: the underutilization of beet by-products. By exploring the production of a phenol-rich powder, the research aligns with the growing trend towards sustainability and waste reduction in the food industry. The potential application of the product as a natural colorant or ingredient adds value, particularly given the increasing consumer demand for natural additives. The study employs advanced analytical techniques, including Infrared spectrometry, X-ray fluorescence, ultra-high performance liquid chromatography (UHPLC), and scanning electron microscopy (SEM). These methods are well-suited for characterizing the chemical and physical properties of the atomized powder, providing a comprehensive understanding of its composition and structure.

General comments:

  • The abstract needs more revision, authors must make clear the real motivation of the study. It is not clear in the abstract. Also, please provide a more detailed, step-by-step description of the methodology used in the work.
  • While the introduction provides a good overview of the current state of knowledge, the discussion could better integrate the findings with existing literature.
  • The selection of spray drying as a method for powder production is appropriate, given its wide application in food processing for producing stable and easy-to-handle powders. However, the choice of temperature (140-160°C) should be carefully justified, considering the thermal sensitivity of phenolic compounds. Comparison study should be done for validation.
  • The study examines key variables: temperature, extract input flow rate, and solvent type that are critical in determining the quality and yield of the final product. The choice of ethanol-water (50/50 v/v) and ethanol as solvents is logical, given their efficacy in extracting phenolic compounds. However, the impact of these solvents on the stability and bioavailability of the phenolic compounds should be further discussed.
  • Polyvinylpyrrolidone (PVP) was used as a carrier agent. While PVP is known for its binding properties, the manuscript notes the presence of SiO2, likely from the carrier agent, which may contribute to caking. This highlights a need for alternative carriers that might minimize this issue, which is crucial for industrial application.
  • The results show a significant range in both total phenols content (136-1026 mg AGA/g) and yield (60-89%), indicating the strong influence of the processing parameters. The high phenolic content, especially at the upper end, suggests that the process can effectively concentrate bioactive compounds, though the variability indicates a need for process optimization.
  • The detection of compounds such as betanin, isobetanin, and neobetanin confirms the retention of bioactive pigments, which is a positive outcome for potential use as a natural colorant. The presence of major oxides (K2O, Al2O3, and SiO2) should be carefully evaluated in terms of their implications for both safety and functionality in food applications.
  • Authors must carefully check the manuscript for spelling and grammatical errors and correct them, as in Figure 1 on page 4: change 'beef' to 'beet' in the figure." etc……

Finally, this research contributes valuable insights into the valorization of beet by-products, offering a potential avenue for the production of high-value food ingredients. The findings are promising, but the study also highlights areas for further exploration to ensure the product's feasibility and stability in industrial applications. The study concludes with a positive outlook on the potential applications of the atomized powder, yet it emphasizes the need for further research. Future work could explore different carrier agents, optimize spray drying conditions to maximize phenolic content while minimizing thermal degradation, and assess the functional properties of the powder in various food matrices.

Author Response

  • Comments 1: The abstract needs more revision, authors must make clear the real motivation of the study. It is not clear in the abstract. Also, please provide a more detailed, step-by-step description of the methodology used in the work.
  • Answer1: Thank you for your valuable revision. The abstract has been modified to highlight the motivation of the study, as a proposal for validation of by-products from the industrialization of beet root. Similarly, the methodology has been modified to improve the abstract comprehension.

  • Comments 2: While the introduction provides a good overview of the current state of knowledge, the discussion could better integrate the findings with existing literature.
  • Answer 2: We agree with this comment. Therefore, in section 1, we updated the text, namely by adding line 49-51 and 72 – 98 and deleting the first two paragraphs.

  • Comments 3: The selection of spray drying as a method for powder production is appropriate, given its wide application in food processing for producing stable and easy-to-handle powders. However, the choice of temperature (140-160°C) should be carefully justified, considering the thermal sensitivity of phenolic compounds. Comparison study should be done for validation.
  • Answer 3: Thank you for pointing this out. Regarding the results on the effect of the variables studied (temperature, flow, and solvent) on the total phenolic content (section 3.2), we observed that temperature did not have a significant effect on the total phenolic content, therefore in our research, the applied temperature range (140 -160°C) did not show a negative effect on the total phenolic content. Additionally, in the optimization of the spray drying process (section 3.3), it was determined that the optimal temperature for the extraction of total phenolic compounds is 160 °C.
  • Comments 4: The study examines key variables: temperature, extract input flow rate, and solvent type that are critical in determining the quality and yield of the final product. The choice of ethanol-water (50/50 v/v) and ethanol as solvents is logical, given their efficacy in extracting phenolic compounds. However, the impact of these solvents on the stability and bioavailability of the phenolic compounds should be further discussed.
  • Answer 4: Thank you for pointing this out. In section 3.3, we updated the text, and added lines 356 – 359 and 377-379.
  • Comments 5: Polyvinylpyrrolidone (PVP) was used as a carrier agent. While PVP is known for its binding properties, the manuscript notes the presence of SiO2, likely from the carrier agent, which may contribute to caking. This highlights a need for alternative carriers that might minimize this issue, which is crucial for industrial application.
  • Answer 5:  We agree with this comment, in fact, as a research group we have worked using maltodextrin and gum arabic as a carrier agent, however, these results have not yet been published in scientific journals, but they are available at the following link: VisorHub (utpl.edu.ec). Additionally, in section 5 (line 633), it was included that an alternative to reduce caking, the use of other encapsulants, as well as separating the sugars from the beet by-product extract.

  • Comments 6:  The results show a significant range in both total phenols content (136-1026 mg AGA/g) and yield (60-89%), indicating the strong influence of the processing parameters. The high phenolic content, especially at the upper end, suggests that the process can effectively concentrate bioactive compounds, though the variability indicates a need for process optimization.
  • Answer 6: Thank you for your comment. In section 3.3, we conducted the optimization of the Extract-Solvent variable, which had a significant influence on both the process yield and the total phenolic content of the encapsulated product (Figure 2). While it is true that the process yield was also influenced by the process flow, due to limitations of the spray dryer equipment available in our laboratory, we were unable to work with flow rates below 10%. It is important to note that the optimal conditions obtained from the spray drying process of Beet By-Products have been added in table 3.
  • Comments 7: The detection of compounds such as betanin, isobetanin, and neobetanin confirms the retention of bioactive pigments, which is a positive outcome for potential use as a natural colorant. The presence of major oxides (K2O, Al2O3, and SiO2) should be carefully evaluated in terms of their implications for both safety and functionality in food applications.
  • Answer 7: We appreciate your suggestion. We have modified the last paragraph of section 3.5.2. The presence of Al2O3, K2O and SiO2, does not represent hazard in health doubt they have low on no intestinal absorption, neither diminish the absorption of antioxidants because they do not represent competition in absorption.
  • Comments 8: Authors must carefully check the manuscript for spelling and grammatical errors and correct them, as in Figure 1 on page 4: change 'beef' to 'beet' in the figure." etc……
  • Answer 8: Thanks for the observation. The authors corrected the grammatical errors in Figure 1. and graphical abstract.

Reviewer 2 Report

Comments and Suggestions for Authors

Introduction:

This section should better identify the research gap that the present study intends to fill since the work is not novel by any means.  

Line 125: Spray drying needs more details for reproducibility: please replace 10% or 30% pump capacity for actual feed flow rate either volumetric or mass; what was the atomization gas flow rate or pressure; What do you mean by changing the aspiration rate, this should change the drying gas flow rate. What was the drying gas flow rate then.

Line 272: please don’t make this comparison - Wheat flour and wheat semolina are different types of food products with distinct physical and chemical properties. The stability requirements for these products, particularly in terms of moisture content, may not directly apply to a dehydrated beet by-product, which has a different composition, structure, and intended use.

Line 310: please replace x1, x2 and x3 with the actual independent factors on the equations. Also put an equation number on each equation.

Line 317 - 321: this is too hard to read

Line 322: what we see in Figure 3b is the same phenolic content for all spray drying conditions.  Please comment.

Is equation “Total Phenolic” the RSM of figure 3b?

Line 336: what data supports this 50/50 is this figure 4? Why 60% of water and not 40%  water?

Figure 3 : you should attribute a value for the extract- solvent of 75 instead of 0;

Line 354 to 362 – this isn’t results or discussion, doesn’t belong here.

Table 2 should be on line 275

Line 367 : I don’t understand, how many samples did you produce on the spray drying for the yield and phenolic data? Why table 2 only present the result for 1 sample?

Section 3.4.1 Color – there is no discussion of results here? Where is the influence of processing conditions on color?

Section 3.4.2 – no discussion here too just results.

Line 410: if you are comparing to other at least comment on why your value is so low. Please note you are presenting the tap density not the bulk density which should be higher.

Susbsequent sections have no discussions, only results. Please provide insights and discuss your results.

Conclusion

Line 544: its not atomization process its spray dryin. And it not optimized, you didn’t use a function to find (desirability for instance) the best yield or phenolic content. If your process was optimized, then you should mention the optimal conditions.

The remaining conclusion is just a summary of results.  

Comments on the Quality of English Language

Line 317 - 321: this is too hard to read

Author Response

  • Comments 1: This section should better identify the research gap that the present study intends to fill since the work is not novel by any means. 
  • Answer 1: We have according modified the introduction (line 49 -51, 70 -98). In the literature review, we found that there is extensive information available on producing powdered products from beet juice or concentrate. However, there is a significant lack of information regarding the use of beet by-products for powder production through spray drying. Therefore, the innovative aspect of our research lies in utilizing these by-products to produce powder.
  • Comments 2: Line 125 - Spray drying needs more details for reproducibility: please replace 10% or 30% pump capacity for actual feed flow rate either volumetric or mass; what was the atomization gas flow rate or pressure; What do you mean by changing the aspiration rate, this should change the drying gas flow rate. What was the drying gas flow rate then.
  • Answer 2: Thank you for your observation. In Section 2.4.2 (line 139), we have included the feed flow rates corresponding to 10% and 30% pump capacity. Regarding your question, “What do you mean by changing the aspiration rate, this should change the drying gas flow rate. What was the drying gas flow rate then?”, the outlet temperature of the process is influenced by the inlet temperature, feed flow rate, and aspiration rate (drying gas flow). In this study, we investigated the effects of temperature and feed flow rate on the yield and total phenol content of the powdered product. To ensure a low outlet temperature, and following preliminary experiments not included in the manuscript, we used the maximum aspirator flow rate of 35 m³/h for all experiments. Consequently, we have removed the paragraph “The outlet temperature was maintained under 50 °C changing the aspiration percentage (Fig 1)” to avoid any confusion.

  • Comments 3: Line 272 - please don’t make this comparison - Wheat flour and wheat semolina are different types of food products with distinct physical and chemical properties. The stability requirements for these products, particularly in terms of moisture content, may not directly apply to a dehydrated beet by-product, which has a different composition, structure, and intended use.
  • Answer3: Agree. We have according removed the paragraph “For products such as wheat flour and wheat semolina, the maximum value allowed is 14.5% moisture, therefore, the dehydrated beet by-product could be considered stable [19,20]”

  • Comments 4: Line 310 - please replace x1, x2 and x3 with the actual independent factors on the equations. Also put an equation number on each equation.
  • Answer 4: The authors appreciate the suggestion. We replaced x1, x2, and x3 with temperature, extract flow, and extract-solvent. Additionally, we numbered each equation (line 334 – 341).

  • Comments 5: Line 317 – 321- this is too hard to read
  • Answer 5: The authors are grateful of the revision. The paragraph has been re write to improve the comprehension (line 334 - 350).

  • Comments 6: Line 322- what we see in Figure 3b is the same phenolic content for all spray drying conditions.  Please comment. Is equation “Total Phenolic” the RSM of figure 3b?
  • Answer 6: Thank you for the question. The authors have replaced Figures 3a and 3b to more clearly visualize the effect of the different variables studied. Given that temperature has the least influence on both yield and total phenol content, an adjustment was made to the graph. The updated version presents the influence of the solvent and flow rate on yield and total phenols, keeping the intermediate temperature value (150 °C) constant.

  • Comments 7: Line 336: what data supports this 50/50 is this figure 4? Why 60% of water and not 40% water?
  • Answer 7: Thank you for your observation, we have included a brief explanation in the manuscript. In fact, the results are similar between these mixtures of ethanol and water, however we selected the 50/50 mixture due to the high content of ethanol make easy the processing as well as helps to prevent the spoilage of the liquid extracts (369 -370, 377). 

  • Comments 8: Figure 3 - you should attribute a value for the extract- solvent of 75 instead of 0;
  • Answer 8: Thank you for the observation. In response to comment 6, the authors have made an adjustment to the Figure 3, so the intermediate value for the extract-solvent variable (0) is no longer shown.

  • Comments 9: Line 354 to 362 – this isn’t results or discussion, doesn’t belong here.
  • Answer 9: Thank you for the revision, the paragraph has been corrected.

  • Comments 10: Table 2 should be on line 275
  • Answer 10: Agree. We placed Table 2 in Section 3.1
  • Comments 11: Line 367 - I don’t understand, how many samples did you produce on the spray drying for the yield and phenolic data? Why table 2 only present the result for 1 sample?
  • Answer 11: Thank you for this observation. The optimization of encapsulation was carried out for encapsulation yield and total phenol content, then the encapsulated produced at the optimized operative conditions of spray-drying was characterized by physical and chemical properties. In the manuscript we have been included this explanation (section 3.4).
  • Comments 12: Section 3.4.1 Color – there is no discussion of results here? Where is the influence of processing conditions on color?
  • Answer 12: This observation could be responded with the answer in the previous comment. Color was measured as characterization of the optimized encapsulated.
  • Comments 13: Section 3.4.2 – no discussion here too just results.
  • Answer 13: The authors have been considered the suggestion and added a comparison and discussion of results to the manuscript in the following sections: 3.4.2 (line 426-429).

  • Comments 14: Line 410: if you are comparing to other at least comment on why your value is so low. Please note you are presenting the tap density not the bulk density which should be higher.
  • Answer 14: The authors considered the suggestion and included the following at the end of Section 3.4.3: “Similarly, in the case of encapsulated powders, the type and percentage of encapsulant used will directly influence the density of the final product”.

  • Commentes 15: Subsequent sections have no discussions, only results. Please provide insights and discuss your results.
  • Answer 15: The authors considered the suggestion and added a comparison and discussion of results to the manuscript in the following sections: 3.4.1 (line 426 -429, )3.4.2 (line 438-445; 456-460), 3.4.3 (line 475 -477, 490 - 492), 3.4.5 (line 454 – 538), 3.5.2 (line 589 -592) and 3.5.2 (line 606 - 614). In addition, a brief remark of the importance of physochemical properties has been added to the manuscript.

  • Comments 16: Line 544- its not atomization process its spray dryin. And it not optimized, you didn’t use a function to find (desirability for instance) the best yield or phenolic content. If your process was optimized, then you should mention the optimal conditions. The remaining conclusion is just a summary of results. 
  • Answer 16: Agree In the conclusion, the following was added: “The extraction solvent factor influenced both the yield and total phenolic content of the encapsulated powder, while the extract flow only affected the total phenolic content. The optimal conditions of the studied factors to maximize yield (temperature, 160 °C; extract flow, 30%; and solvent composition, 50% ethanol) and phenolic content (temperature, 160 °C; extract flow, 10%; and solvent composition, 50% ethanol) in the encapsulated powder were successfully defined.”  And line 632 and 642. Additionally, we replaced the term atomization process with 'spray drying.

Reviewer 3 Report

Comments and Suggestions for Authors

In the introduction section, the related work section of recent years needs to compare the analysis findings of the rather insufficient literatures, the yields of antioxidant and polyphenol amounts and to emphasize the contribution of these research results to the literature

-Please indicate the time period "hours" in the manuscript by shortening it with "h"

-Ultrasonic mixer in the extraction process is a process that increases the extraction yield considerably, I prefer it to classical extraction

-Please refer to the literature for the equations and equations used

-Water activity parameter provides quite decisive ideas especially in shelf life analyses, please add a table comparing the results obtained in this research with the literature

-Readability of graphics and figures is difficult, please increase the font sizes

-Ultrasound is used as a solvent in this research, not methanol, it is a detail that is preferred in terms of health, please emphasize this importance when comparing with the literature.

"According to Fernando et al. [23], the extraction of 322 antioxidant compounds is improved when ethanol or methanol is mixed with water, this 323 occurs due to the polarity of the main beetroot compounds such as betalains. In addition, 324 when ethanol and water are used as solvent the obtained products could be used as a 325 natural colorant in the food industry."

-detailed discussion of the results with encapsulation efficiency is needed

-there are no suggestions for further research in the conclusion section, please add the missing ones

the figures really need correction, they are never readable, especially the spectra...

Encapsulation and antibacterial studies of goji berry and garlic extract in the biodegradable chitosan

G Baysal, HS Olcay, Ç Günneç

Journal of Bioactive and Compatible Polymers 38 (3), 209-219

Author Response

  • Comments 1: In the introduction section, the related work section of recent years needs to compare the analysis findings of the rather insufficient literatures, the yields of antioxidant and polyphenol amounts and to emphasize the contribution of these research results to the literature.
  • Answer 1: We agree with this comment. Therefore, in section 1, we updated the text, namely by adding line 49-51 and 72 – 98 and deleting the first two paragraphs.

  • Comments 2: Please indicate the time period "hours" in the manuscript by shortening it with "h"
  • Answer 2: Agree, we have made the requested changes throughout the manuscript.

  • Comments 3: Ultrasonic mixer in the extraction process is a process that increases the extraction yield considerably, I prefer it to classical extraction
  • Answer 3: Thanks for your observation. We have included an explanation of selecting dynamic maceration in the manuscript, in section 3.3 (line 379).

  • Comments 4: -Please refer to the literature for the equations and equations used
  • Answer 4: In section 2.7, the authors who proposed the use of equation 7 were added. In the other methodologies, the bibliographic references are already included

  • Comments 5: Water activity parameter provides quite decisive ideas especially in shelf life analyses, please add a table comparing the results obtained in this research with the literature
  • Answer 5: We have added a comparison of results with other studies related to beetroot juice powder obtained through spray drying (Section 3.4.2).

  • Comments 6: Readability of graphics and figures is difficult, please increase the font sizes
  • Answer 6: Agree. We increase the font size in all figures.

  • Comments 7: Ultrasound is used as a solvent in this research, not methanol, it is a detail that is preferred in terms of health, please emphasize this importance when comparing with the literature.
  • Answer 7: Thanks for your comment. We have marked the importance of ethanol in the manuscript, in section 3.3.

  • Comments 8: "According to Fernando et al. [23], the extraction of 322 antioxidant compounds is improved when ethanol or methanol is mixed with water, this occurs due to the polarity of the main beetroot compounds such as betalains. In addition, when ethanol and water are used as solvent the obtained products could be used as a 325 natural colorant in the food industry."
  • Answer 8: This observation would be included with the changes made for the previous comment(line 356).

  • Comments 9: -detailed discussion of the results with encapsulation efficiency is needed
  • Answer 9: Thanks for your suggestion. We have included a brief discussion about encapsulation efficiency.

  • Comments 10: there are no suggestions for further research in the conclusion section, please add the missing ones
  • Answer 10: Agree, in the conclusions, a paragraph was added with possible suggestions for future research (line 642 -648).

  • Comments 10: the figures really need correction, they are never readable, especially the spectra...
  • Answer 10: Agree, we improve all figures.

Round 2

Reviewer 1 Report

Comments and Suggestions for Authors

The authors significantly revised the manuscript, corrected all comments, and gave a substantiated answer to all my questions. I think that the article has improved significantly and can be accepted for publication.

Reviewer 2 Report

Comments and Suggestions for Authors

The authors answer all comments. Congratulations. 

Reviewer 3 Report

Comments and Suggestions for Authors

I am pleased to inform accepted of revision manuscript